# MECHANISTIC INTERPRETABILITY OF IN-CONTEXT LEARNING GENERALIZATION THROUGH STRUCTURED TASK CURRICULUM

## ABSTRACT

We study the generalization mechanisms of In-Context Learning (ICL) in Transformer-based language models from the perspective of mechanistic interpretability. While prior works have explored ICL either through single-task mechanistic analysis or multi-task empirical evaluation, a unified mechanistic understanding of ICL generalization has not yet been established. To address this gap, we conduct a systematic study using two structured tasks: ICL Markov Chain and ICL Regression. These tasks respectively instantiate Markovian and i.i.d. data distributions, enabling tractable analysis and principled definitions of task similarity. We investigate how knowledge acquired from a source task facilitates learning in a target task, and how this transfer depends on formal measures of similarity between the tasks. Our empirical results show that initializing from data-dependent checkpoints trained on simpler source tasks significantly improves data efficiency when training on more complex target tasks. Furthermore, we identify mathematically and mechanistically interpretable "common structures" in the Transformer QK circuits at lower layers, along with higher-layer features, that support cross-task generalization. These structures vary predictably with formal similarity metrics. Our work unifies mechanistic and generalization perspectives on ICL, offering new insights into curriculum learning phenomena and informing the design of more scalable and data-efficient ICL training pipelines.

## 1 INTRODUCTION

In-Context Learning (ICL) represents one of the most striking emergent capabilities of large language models, enabling them to rapidly adapt to new tasks using only a handful of input-output examples without updating their parameters (Brown et al., 2020). This phenomenon has transformed how we deploy language models in practice, yet our understanding of the underlying mechanisms remains fragmented. While empirical studies have catalogued ICL's impressive breadth across tasks ranging from arithmetic to commonsense reasoning (Wei et al., 2022), and mechanistic interpretability work has identified specific circuits like induction heads (Olsson et al., 2022), these perspectives have developed in parallel with limited cross-pollination. This disconnect leaves fundamental questions unanswered: How do models transfer ICL capabilities between related tasks? What computational structures enable efficient generalization? And how can we leverage these insights to design better training curricula?

The challenge of understanding ICL generalization lies at the intersection of two research traditions. On one hand, the mechanistic interpretability community has made significant progress in reverse-engineering the specific circuits that enable ICL on individual tasks. For instance, Nichani et al. (2024) and Chen et al. (2024b) demonstrated how transformers implement Markov chain inference through specialized attention patterns, while Chen et al. (2024a) revealed the kernel regression mechanisms underlying in-context linear regression. On the other hand, the empirical machine learning community has shown that models trained on diverse task mixtures exhibit superior ICL capabilities (Kim et al., 2025), suggesting that some form of transfer learning occurs. However, these empirical observations lack mechanistic grounding—we observe that transfer happens but not how or why.

This gap matters for both theoretical and practical reasons. Theoretically, understanding how transformers generalize ICL mechanisms would illuminate fundamental questions about the nature of few-shot learning and the inductive biases of attention-based architectures. Practically, such understanding could revolutionize how we train language models. Current pretraining requires massive computational resources partly because we lack principled ways to sequence tasks for efficient learning. If we understood which tasks share computational structures and how these structures transfer, we could design curricula that achieve comparable capabilities with significantly less data and compute.

**Our Approach.** We bridge the mechanistic and empirical perspectives through controlled experiments on structured ICL tasks. We focus on two complementary task families that capture distinct data regimes: (i) Markov chains, which model sequential dependencies where each element depends on its predecessors, and (ii) linear regression, which models independent and identically distributed (i.i.d.) relationships between inputs and outputs. These tasks serve as tractable models of real-world ICL scenarios while remaining amenable to mechanistic analysis. Crucially, both tasks admit natural complexity hierarchies—Markov chains vary in order and dependency structure, while regression tasks vary in their degree of nonlinearity—allowing us to study transfer between simpler and more complex variants.

Our experimental framework follows the standard pretraining-finetuning paradigm but with careful mechanistic monitoring. We first pretrain transformers on simpler task variants (e.g., 2-gram Markov chains or linear regression), then fine-tune on more complex variants (e.g., 3-gram Markov chains or quadratic regression). Throughout training, we track the evolution of specific model components—attention patterns, QK circuits, and feed-forward networks—to identify which structures remain stable (suggesting reuse) versus which adapt (suggesting task-specific learning). This approach reveals not just that transfer occurs, but precisely which computational mechanisms enable it.

**Key Contributions.** Our investigation yields three main contributions:

- **Mechanistic characterization of ICL transfer:** We identify specific "common structures" that enable generalization across tasks. In Markov chains, the induction circuit learned for 2-gram inference remains largely intact when adapting to 3-gram, with only the causal structure embeddings requiring updates. In regression tasks, the data-copying attention patterns provide stable scaffolding for learning increasingly nonlinear relationships. These findings provide the first mechanistic account of how ICL capabilities transfer between related tasks.
- **Quantitative efficiency gains from curriculum learning:** Our experiments demonstrate that appropriate pretraining reduces data requirements by up to 10x compared to training from scratch. For instance, models pretrained on linear regression can learn quadratic regression with 90% fewer examples, while direct training on quadratic regression often fails entirely within the same budget. Moreover, we show that transfer efficiency correlates with formal task similarity metrics, providing principled guidelines for curriculum design.
- **Practical implications for LLM training:** Our findings suggest that the common practice of training on diverse tasks simultaneously may be suboptimal. Instead, our results advocate for structured curricula that sequence tasks based on shared computational requirements. We provide concrete recommendations for task ordering based on mechanistic similarity rather than surface-level task categories.

**Paper Organization.** The remainder of this paper is organized as follows. Section 2 establishes our experimental framework, introducing the transformer architecture and our two structured task families. Section 3 presents our findings on Markov chain tasks, revealing how induction circuits enable transfer across different chain orders. Section 4 examines regression tasks, demonstrating the critical role of data-copying mechanisms in learning nonlinear functions. Section 5 concludes with implications for both ICL theory and practical training strategies. Technical details and additional experiments are provided in the appendices.

## 1.1 RELATED WORKS

**Mechanistic Interpretability of ICL.** Recent work has made significant progress in understanding the circuits underlying in-context learning. Olsson et al. (2022) identified induction heads as a key mechanism for copying patterns from context, while Elhage et al. (2021) developed the mathematical framework for analyzing attention patterns. For specific tasks, Nichani et al. (2024) and Chen et al. (2024b) characterized how transformers implement Markov chain inference, Chen et al. (2024a) and He et al. (2025) analyzed in-context linear regression, and Cabannes et al. (2024) studied algorithmic tasks. However, these works focus on individual tasks in isolation, leaving the question of cross-task transfer unexplored.

**Task Diversity and Transfer in ICL.** Empirical studies have shown that training on diverse tasks improves ICL performance (Kim et al., 2025; Zhang et al., 2022), with some work suggesting the existence of "common structures" that facilitate transfer. However, these studies lack mechanistic grounding and rely on black-box performance metrics. Our work bridges this gap by providing mechanistic evidence for these common structures and characterizing their role in transfer.

**Curriculum Learning.** The broader curriculum learning literature (Bengio et al., 2009; Xu et al., 2020; Edelman et al., 2024; Graves et al., 2017) has long advocated for training on progressively harder tasks, with recent work applying these ideas to language models (Soviany et al., 2022; Jia et al., 2022; Naïr et al., 2024; Lin et al., 2024; Feng et al., 2025). Our contribution is to provide mechanistic justification for curriculum design in the specific context of ICL, showing which task orderings lead to successful transfer and why.

**Notations.** For a sequence $x = (x_1, x_2, \cdots, x_N)$, $x_{i:j}$ denotes its subsequence $(x_i, x_{i+1}, \cdots, x_j)$; for a positive integer $N$, $[N]$ denotes a set $\{1, 2, \cdots, N\}$; for a sample space $\mathcal{S}$, $\Delta_{\mathcal{S}}$ denotes the probability simplex on $\mathcal{S}$.

## 2 SETUP

### 2.1 TRANSFORMER ARCHITECTURE

We consider a standard transformer architecture, as described in prior work (Nichani et al., 2024; Chen et al., 2024b; Edelman et al., 2024). A transformer processes any input sequence $X \in \mathbb{R}^{L \times D}$, where $L$ is the sequence length and $D$ is the embedding dimension. Each column $X_{:,i}$ represents the embedding for the $i$-th token. The model applies a series of blocks, each containing an attention layer followed by a feed-forward network (FFN). These two layers, defined in (B.1) and (B.2), are composed sequentially to form a transformer block, and the full model is a stack of these blocks.

**Definition 2.1** (Transformer). *An $N$-layer decoder-only transformer, denoted, $\text{TF}_\theta(\cdot)$, is the composition of $N$ blocks. Given an input embedding $h^{(0)} = X \in \mathbb{R}^{L \times D}$ and unembedding matrix $W_o \in \mathbb{R}^{D \times D_o}$, the transformer output is given by*

$$\text{TF}_\theta(h^{(0)}) = h^{(N)} W_o \in \mathbb{R}^{L \times D_o}, \qquad h^{(\ell)} = \text{FFN}_{\theta_{\text{FFN}}} \circ \text{Attn}_{\theta_{\text{Attn}}}\left(h^{(\ell-1)}\right) \in \mathbb{R}^{L \times D}, \qquad \forall \ell \in [N],$$
(2.1)

*where the parameter $\theta = \{\theta_{\text{Attn}}^{(1:N)}, \theta_{\text{FFN}}^{(1:N)}\}$ consists of both the attention layers $\theta_{\text{Attn}}^{(\ell)} = \{Q_h^{(\ell)}, K_h^{(\ell)}, V_h^{(\ell)}, O_h^{(\ell)}\}_{h=1}^H \subset \mathbb{R}^{D \times D}$ and the FFN layers $\theta_{\text{FFN}}^{(\ell)} = \{W_1^{(\ell)}, W_2^{(\ell)}\} \subset \mathbb{R}^{D \times D}$.*

Beyond the standard transformer architecture with both attention layer and FFN layer, we will also consider attention-only transformers, i.e., $W_1^{(\ell)}, W_2^{(\ell)} = \mathbf{0}$ for all $\ell \in [N]$.

**QK Circuit and OV Circuit.** The attention mechanism's computation in (B.1) involves two stages: first, forming attention patterns via query-key interactions, and second, writing information to the output via value-vector aggregation. Following Elhage et al. (2021), we refer to the composite matrices governing these stages as the Query-Key (QK) circuit, defined as $W_{\text{QK},h}^{(\ell)} := Q_h^{(\ell)} K_h^{(\ell)^\top}$, and the Output-Value (OV) circuit, defined as $W_{\text{OV},h}^{(\ell)} := V_h^{(\ell)} O_h^{(\ell)}$.

## 2.2 STRUCTURED TASK DESCRIPTION: IN-CONTEXT MARKOV CHAIN AND LINEAR REGRESSION

We adapt two well-studied in-context learning (ICL) tasks from the literature to analyze model performance on distinct data structures. Specifically, we use an ICL Markov chain task to model sequentially dependent (Markovian) data (Nichani et al., 2024; Chen et al., 2024b; Edelman et al., 2024; Zhou et al., 2024) and an ICL linear regression task to model independent and identically distributed (i.i.d.) data (Chen et al., 2024a; He et al., 2025; Akyürek et al., 2023; Zhang et al., 2024).

### 2.2.1 IN-CONTEXT MARKOV CHAIN

**Data Generation for $n$-Gram Markov Chains.** Standard language modeling task predicts the next token given all previous tokens: $x_{i+1} \sim p(\cdot|x_{1:i})$, while $n$-gram modeling simplifies this by giving only $(n-1)$ previous tokens for prediction. Our In-Context Markov Chain task is derived from $n$-gram modeling, which is defined as follows:

**Task 2.2** ($n$-Gram Markov Chain). *We assume the data sequence is generated from an $n$-gram Markov chain model denoted by a 4-tuple: $(\mathcal{S}, n, \mathtt{pa}, \mathsf{P}_\pi)$, where $\mathcal{S} = \{1, 2, \cdots, S\}$ is a finite corpus and $n$ is the $n$-gram coefficient. Given a token position $i$, $\mathtt{pa}(i)$ denotes the set of parents of $i$, with $|\mathtt{pa}(i)| = n-1$ or $0$. We define the set of root nodes $\mathcal{R} = \{i : \mathtt{pa}(i) = \emptyset\}$. For $|\mathtt{pa}(i)| = n-1$, we write multiset $\mathtt{pa}(i)$ as $\mathtt{pa}(i) = \{\mathtt{pa}_1(i), \mathtt{pa}_2(i), \cdots, \mathtt{pa}_{n-1}(i)\}$, where $1 \leq \mathtt{pa}_1(i) \leq \mathtt{pa}_2(i) \leq \cdots \leq \mathtt{pa}_{n-1}(i) \leq i-1$, and call $\mathtt{pa}_j(i)$ the $j$-th parent of $i$ with $1 \leq j \leq n-1$. We also let $\mathsf{P}_\pi$ denote the prior probability distribution over a set of Markovian transition dynamics specified by parent structure defined by $\mathtt{pa}$. We then generate the data sequence as follows: (i) sample $\pi \sim \mathsf{P}_\pi$, where $\pi : \mathcal{S}^{n-1} \to \Delta_\mathcal{S}$, and calculate the unique stationary distribution $\mu_\pi$ of $\pi$, where $\mu_\pi \in \Delta_\mathcal{S}$, (ii) for $i = 1, 2, \cdots, L$, sample $s_i \overset{i.i.d.}{\sim} \mu_\pi$ if $i \in \mathcal{R}$, otherwise sample $s_i \sim \pi(\cdot|\mathtt{pa}_{1:(n-1)}(i))$, (iii) for $i = L+1, L+2, \cdots, L+(n-1)$, sample $s_i \overset{i.i.d.}{\sim} \mathsf{Unif}(\mathcal{S})$ and sample $s_{L+n} \sim \pi(\cdot|s_{L+1:L+(n-1)})$, (iv) return the data $x = s_{1:L+(n-1)}$ and label $y = s_{L+n}$.*

In our experiments, we set $s_{1:n-1}$ as root nodes and $s_{n:L}$ as non-root nodes. Subsequence $s_{1:L}$ serves as the context of the latent interpositional causal structure specified by $\mathtt{pa}$ and $\pi$. Besides, subsequence $s_{L+1:L+(n-1)}$ are queries independent to the Markovian transition dynamics $\pi$, and label $s_{L+n}$ is derived from the queries through transition $\pi$. The model is expected to predict the label $s_{L+n}$ given the queries $s_{L+1:L+(n-1)}$ and the context $s_{1:L}$.

**From Parent Structure to Causal Matrix.** Given the $p$-th parent multiset $\mathtt{pa}_p$ with $p \in [n-1]$, we define the corresponding $p$-th causal matrix $M^p \in \mathbb{R}^{L \times L}$ as follows: entry $M^p_{i,j} = 1$ when $j = \mathtt{pa}_p(i)$, otherwise $M^p_{i,j} = 0$. If we want to look up the $p$-th parent of non-root node $i$, we can consider the $i$-th row of $M^p$, i.e. $M^p_{i,:}$, and there is only one non-zero entry in this row, which is exactly $M^p_{i,\mathtt{pa}_p(i)}$.

**Embedding, Readout and Training Objective.** We take 3 steps to embed the data sequence: (1) zero padding, aligning the length of data sequence parameterized by different $n$; (2) one-hot token embedding, embedding the data sequence into one-hot vectors; (3) one-hot position embedding, concatenating one-hot vectors to the token-embedded sequence. Moreover, we read out the model prediction at the last query position, i.e. the embedded position of $s_{L+(n-1)}$ defined in Task 2.2. For training setting, we use attention-only transformer model and set the output dimension $D_o = S$. We adopt the cross-entropy (CE) loss and formulate the training objective as:

$$\mathcal{L}_{\mathrm{CE}}(\theta) = \mathbb{E}_{\pi \sim \mathsf{P}_\pi, \, s_{1:L+n} \sim \mathsf{P}(\cdot|\mathtt{pa}, \pi)}\big[-\log\big(\mathtt{smax} \circ \mathtt{read} \circ \mathtt{TF}_\theta \circ \mathtt{emb}(s_{1:L+(n-1)})_{s_{L+n}}\big)\big],$$

where $\mathsf{P}(\cdot|\mathtt{pa}, \pi)$ denotes the joint distribution of Markov chain trajectory under parent structure $\mathtt{pa}$ and transition rule $\pi$. Here, $\mathtt{emb}$ and $\mathtt{read}$ denote the data embedding and readout operators.

### 2.2.2 IN-CONTEXT LINEAR REGRESSION

**Data Generation for In-Context Linear Regression.** The in-context linear regression task requires the model to perform regression on a dataset $\{(x_i, y_i)\}_{i=1}^N$ that is provided within the context. This in-context dataset is generated as follows:

**Task 2.3** (Linear Regression). *We assume the data sequence is generated from a link function $\varsigma$ : $\mathbb{R}^d \to \mathbb{R}$ in the following steps: (i) sample a parameter vector $\beta \sim \mathcal{N}(\mathbf{0}, I_d)$, where $\beta \in \mathbb{R}^d$; (ii) for each $i \in [N+1]$, we sample $x_i \overset{i.i.d.}{\sim} \mathcal{N}(\mathbf{0}, I_d)$ with $x_i \in \mathbb{R}^d$, and compute the corresponding output $y_i = \varsigma(\beta^\top x_i) \in \mathbb{R}$; (iii) return the data $x = (x_{1:N+1}, y_{1:N})$ and label $y = y_{N+1}$.*

Here, the data domain $(x_{1:N}, y_{1:N})$ serves as the context of the latent nonlinear mapping $\varsigma$, and the token $x_{N+1}$ serves as the query. The model is expected to predict the label $y_{N+1}$ given the context $(x_{1:N}, y_{1:N})$ and the query $x_{N+1}$.

**Embedding, Readout and Training Objective.** We embed data in 2 steps: (1) dimension alignment, aligning the dimension of the context label $y_i \in \mathbb{R}$ and input $x_i \in \mathbb{R}^d$ by padding zeros to the end of each $y_i$; (2) position rearrangement, rearranging the position of data $x = (x_{1:N+1}, y_{1:N})$ to $x = (x_1, y_1, x_2, y_2, \cdots, x_N, y_N, x_{N+1})$, where the query $x_{N+1}$ is moved to the last token position and each context label $y_i$ immediately follows its corresponding input $x_i$; (3) one-hot position embedding, concatenating one-hot vectors to the positionally formatted sequence. Moreover, we read out the model prediction at the last token position.

For training setting, We use standard transformer model and set the output dimension $D_o = 1$. We select the mean squared error (MSE) loss and formulate the training objective as:

$$\mathcal{L}_{\text{MSE}}(\theta) = \mathbb{E}_{\beta \sim \mathcal{N}(\mathbf{0}, I), (x_i, y_i)_{i=1}^N \overset{i.i.d.}{\sim} \mathsf{P}(\cdot | \varsigma, \beta)} \left[ \left( \texttt{read} \circ \texttt{TF}_\theta \circ \texttt{emb}(x_{1:N+1}, y_{1:N}) - y_{N+1} \right)^2 \right],$$

where $\mathsf{P}(\cdot | \varsigma, \beta)$ denotes the joint distribution of any input-output pair $(x_i, y_i)$ under link function $\varsigma$ and parameter vector $\beta$. Here, $\texttt{emb}$ and $\texttt{read}$ denote the data embedding and readout operators.

### 2.3 PRE-TRAINING AND SUPERVISED FINE-TUNING

Standard training for large language models (LLMs) typically involves two sequential stages. First, a model undergoes **pretraining** on a massive, general-domain dataset to acquire broad linguistic competence and world knowledge. Second, the model is adapted through **supervised finetuning (SFT)** on a smaller, specialized dataset to learn task-specific expertise (e.g., Devlin et al., 2019; Brown et al., 2020). While these two stages often share the same architecture and training objective, they operate on distinct data distributions. It is well-observed that the SFT stage is significantly more data-efficient than pretraining. To fully understand the benefits of pretraining and the efficiency of SFT, it remains important to answer the following question:

*How do the model's parameters change from their pretrained initialization during the SFT stage?*

In our experiments, we pretrain and then finetune a transformer model on different variations of the same structured task, i.e., n-gram Markov chain and linear regression. These variations, which we term **subtasks**, are generated by using distinct hyperparameter settings. For the Markov chain, a subtask is defined by its order and transition probabilities $(n, \texttt{pa})$; for linear regression, it is defined by the link function $\varsigma$. In this framework, we conceptualize the general structured task as "**common knowledge**" and a specific subtask as "**expertise knowledge**". This allows us to seek a clear mechanistic interpretation for the following central questions:

*(i) Does the "common knowledge" learned during pretraining accelerate adaptation to the "expertise" required by the SFT subtask? (ii) How do the model's internal components evolve during this transition?*

## 3 STUDY 1: IN-CONTEXT MARKOV CHAIN

In this section, we conduct pretraining and finetuning experiments on the $n$-Gram Markov Chain task (Task 2.2) using a 2-layer, 2-head attention-only transformer. We design three subtasks by varying two factors: the order $n$ and the parent structure $\texttt{pa}$ of each token, which is denoted by a 2-tuple $(n, \texttt{pa})$. The three subtasks are listed below: (i) $(2, \texttt{diag})$: 2-gram with only one parent $\texttt{pa}(i) = i - 1$; (ii) $(3, \texttt{diag})$: 3-gram with two parents $\texttt{pa}_1(i) = i - 1, \texttt{pa}_2(i) = i - 2$; (iii) $(2, \texttt{free})$: 2-gram with only one randomly generated parent $\texttt{pa}_1$, details of which are deferred to Appendix D.1. For all experiments, we use a vocabulary of $\mathcal{S} = \{1, 2, 3\}$ and set the prior for the transition probabilities to a Dirichlet distribution $\mathsf{P}_\pi = \text{Dir}(0.1 \cdot \mathbf{1})$.

### 3.1 PRETRAINING STAGE: TRANSFORMER LEARNS INDUCTION HEAD

Previous work (Nichani et al., 2024; Chen et al., 2024b; Olsson et al., 2022) has shown that a two-layer, $(n-1)$-head attention-only transformer can solve the $n$-Gram Markov Chain by implementing an *induction head* mechanism. In this paper, we first restate this mechanism and then present detailed experiments designed to empirically validate its functionality.

> **Mechanism 1 (Induction Head).** This mechanism operates across two layers to solve the task:
>
> - **Layer 1: Causal Structure Embedding.** The first layer is responsible for *embedding the chain's causal structure*. Each attention head $p \in [n-1]$, is dedicated to a specific parent position, and its QK circuit $W_{\mathsf{QK},p}^{(1)}$ learns to encode the $p$-th causal matrix $M^p$.
> - **Layer 2: Induction Circuit.** The second layer functions as an *induction circuit* to estimate the empirical transition. It scans the parents of the whole context for all that equal to the query, and aggregates the corresponding child nodes to build an empirical probability distribution for the next token. Recall that $\mathcal{R} = \{i : \mathtt{pa}(i) = \emptyset\}$, then such process can be formulated as:
>
> $$\mathsf{TF}_\theta(s|s_{L+1:L+(n-1)}) \approx \frac{\sum_{j \notin \mathcal{R}} \mathbb{1}(\mathtt{pa}_{1:(n-1)}(j) = s_{L+1:L+(n-1)}, \; j = s)}{\sum_{j \notin \mathcal{R}} \mathbb{1}(\mathtt{pa}_{1:(n-1)}(j) = s_{L+1:L+(n-1)})}, \qquad \forall s \in \mathcal{S}. \tag{3.1}$$

**Mechanism Verification.** To verify causal structure embedding at layer 1, we visualize $W_{\mathsf{QK}}^{(1)}$ under setup $(2, \mathtt{free})$ in Figure 1 (i) and observe $W_{\mathsf{QK},1}^{(1)}$ encodes the causal matrix $M^1$. To verify induction circuit at layer 2, we visualize the distribution of cosine similarity between the output of empirical estimator Eq. (3.1) and a $(2, \mathtt{free})$ probing model, whose $W_{\mathsf{QK},1}^{(1)}$ is replaced by the ground-truth causal matrix $M^1$ so as to eliminate the influence from layer 1 model weights. The distribution is visualized in Figure 1 (ii), with the probability density clusters around 1.0.

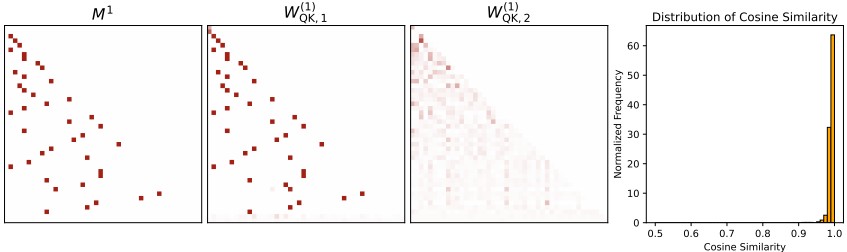

Figure 1: Visualization of Induction Head mechanism under setup $(2, \mathtt{free})$. **(i) Left Heatmaps:** $W_{\mathsf{QK}}^{(1)}$ **embeds the causal structure.** The 3 heatmaps from left to right represent the causal matrix $M^1$ and 2 heads of layer 1 QK circuits, i.e. $W_{\mathsf{QK},1}^{(1)}$ and $W_{\mathsf{QK},2}^{(1)}$. $W_{\mathsf{QK},1}^{(1)}$ encodes the causal matrix $M^1$, while $W_{\mathsf{QK},2}^{(1)}$ remains dummy since 2-gram setting has only one parent. **(ii) Right Histogram: Layer** 2 **learns the induction circuit.** The histogram displays the distribution of cosine similarity between the empirical estimator Eq. (3.1) and the $(2, \mathtt{free})$ probing model. The probability density clusters around 1.0, demonstrating the similarity between layer 2 and the empirical estimator.

### 3.2 SFT STAGE: THE LEARNED CAUSAL STRUCTURE AND INDUCTION CIRCUIT FOSTER GENERALIZATION

We conduct two SFT experiments from 2-gram to 3-gram: $(2, \mathtt{diag}) \to (3, \mathtt{diag})$ and $(2, \mathtt{free}) \to (3, \mathtt{diag})$, whose results are compared to the pretraining controlled group $(3, \mathtt{diag})$. To examine generalization efficacy, we apply 3 metrics: test loss, test accuracy, and test KL-divergence (between output of the model and empirical estimator Eq. (3.1)). Figure 2 (i) displays successful generalization results of the 2 SFT checkpoints. In terms of generalization mechanism, we summarize the following Takeaways:

**Takeaway 1: Unchanged Induction Circuit Fosters Learning of New Causal Structure.** 2-gram pretraining checkpoint learns main structure of induction circuit, which remains unchanged throughout 3-gram SFT and thereby fosters learning of new causal structure.

**Experiment Design.** We make comparisons between $(2, \texttt{free}) \rightarrow (3, \texttt{diag})$ and $(3, \texttt{diag})$ around 2 claims: (i) **Layer** 2 **induction circuit remains unchanged.** We first probe a $(2, \texttt{free})$ model by replacing its $W_{\mathsf{QK}}^{(1)}$ with $(3, \texttt{diag})$ causal matrices, and then SFT it to $(3, \texttt{diag})$ and compare the generalization evaluation results to other checkpoints. Given the generalized layer 1 causal structure, if the main structure of layer 2 induction circuit is already learned in pretraining, the generalization should be super fast as there is little to adjust; (ii) $W_{\mathsf{QK}}^{(1)}$ **captures new causal structure.** We visualize $W_{\mathsf{QK}}^{(1)}$ of the checkpoints to examine whether they captures new $(3, \texttt{diag})$ causal matrices.

**Experiment Results.** For claim (i), we observe from Figure 2 (i) that the $(2, \texttt{free})$ probing model generalizes within 50 ($< 2.5\%$ total) training steps, which is $10\times$ faster than $(2, \texttt{free}) \rightarrow (3, \texttt{diag})$ checkpoint. For claim (ii), we observe from Figure 2 (ii) that all SFT checkpoints captures the $(3, \texttt{diag})$ causal structure pattern, with $W_{\mathsf{QK},1}^{(1)}$ and $W_{\mathsf{QK},2}^{(1)}$ encode $\texttt{pa}_1(i) = i - 1$ and $\texttt{pa}_2(i) = i - 2$ parent structure, respectively.

**Takeaway 2: Shared Causal Structure Further Accelerates and Deepens Generalization.** A shared parent structure $\texttt{pa}_1$ between 2- and 3-gram setting provides distributional mutual information, which deepens learning of another parent structure $\texttt{pa}_2$ and further accelerates generalization compared to SFT between irrelevant causal structure.

**Experiment Design.** We make comparisons between $(2, \texttt{diag}) \rightarrow (3, \texttt{diag})$ and $(2, \texttt{free}) \rightarrow (3, \texttt{diag})$ around 2 claims: (i) **Generalization of** $(2, \texttt{diag}) \rightarrow (3, \texttt{diag})$ **is faster and deeper than** $(2, \texttt{free}) \rightarrow (3, \texttt{diag})$**.** We compare the generalization evaluation results between checkpoints to examine whether $(2, \texttt{free}) \rightarrow (3, \texttt{diag})$ checkpoint generalizes faster and deeper; (ii) **Shared causal structure** $\texttt{pa}_1$ **deepens learning of** $\texttt{pa}_2$**.** We visualize $W_{\mathsf{QK}}^{(1)}$ of the checkpoints to examine the precision of the causal structure they capture.

**Experiment Results.** For claim (i), we observe from Figure 2 (i) that $(2, \texttt{diag}) \rightarrow (3, \texttt{diag})$ generalizes $2\times$ faster and achieves lower convergence test KL-divergence than $(2, \texttt{free}) \rightarrow (3, \texttt{diag})$. For claim (ii), we observe from Figure 2 (ii) that only $(2, \texttt{diag}) \rightarrow (3, \texttt{diag})$ captures the complete $(3, \texttt{diag})$ causal matrices, while $(2, \texttt{free}) \rightarrow (3, \texttt{diag})$ learns some errors and flaws.

## 4 STUDY 2: IN-CONTEXT LINEAR REGRESSION

We study Linear Regression (Task 2.3) using a 2-layer single head standard transformer. Here, we consider the following 3 subtask setups: (i) Lin: linear link function, $\varsigma(x) = \beta^\top x$; (ii) Quad: quadratic link function, $\varsigma(x) = (\beta^\top x)^2$; (iii) Cube: cubic link function, $\varsigma(x) = (\beta^\top x)^3$. Without ambiguity, we slightly abuse notation by defining $W_{\mathsf{QK},\mathsf{pos}}^{(\ell)}$ as the position embedding domain of $W_{\mathsf{QK}}^{(\ell)}$.

### 4.1 PRETRAINING STAGE: TRANSFORMER COPIES DATA VECTOR TO LABEL POSITIONS

Previous works (Chen et al. (2024a); He et al. (2025)) have demonstrated that a single layer multi-head attention-only transformer applies kernel regression mechanism to solve Linear Regression (Task 2.3) with data and label coupled in the same token position. We supplement this result by considering more general data-label decoupling setting and find that transformer implements *copying head* mechanism at layer 1 attention block to couple the corresponding data and label:

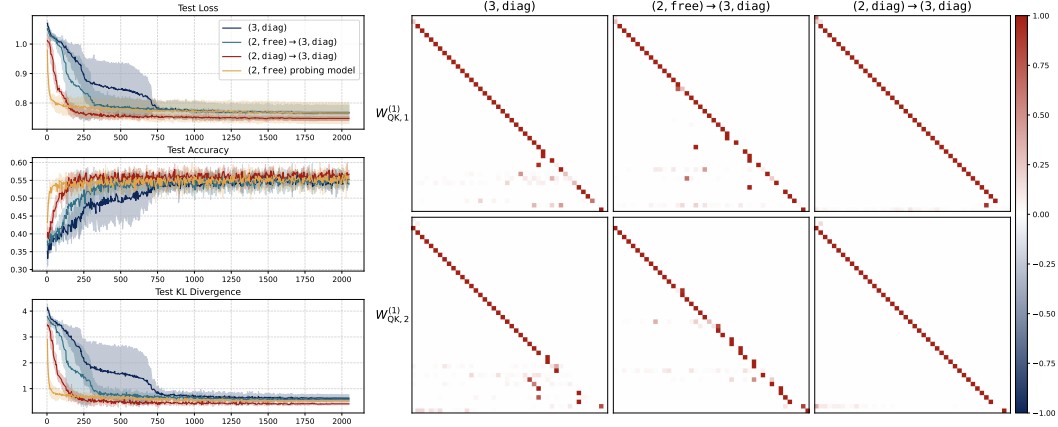

Figure 2: Visualization of ICL Markov Chain SFT results. **(i) Left Plots: Generalization evaluations.** The 3 plots from top to down represent test loss, test accuracy, and test KL-divergence, where formal definitions are deferred to Appendix C. Among all plots, all SFT checkpoints reduce the training plateau at 250-1000 training steps and achieve comparable convergence value compared to $(3, \mathtt{diag})$; **(ii) Right Heatmaps:** $W_{\mathrm{QK}}^{(1)}$ **of SFT checkpoints.** The 6 heatmaps display $W_{\mathrm{QK}}^{(1)}$ of pre-training controlled group $(3, \mathtt{diag})$ (the left column), SFT checkpoints $(2, \mathtt{free}) \rightarrow (3, \mathtt{diag})$ (the middle column), and $(2, \mathtt{diag}) \rightarrow (3, \mathtt{diag})$ (the right column).The top and bottom rows represent $W_{\mathrm{QK},1}^{(1)}$ and $W_{\mathrm{QK},2}^{(1)}$, which capture the $(3, \mathtt{diag})$ parent structure $\mathtt{pa}_1(i) = i - 1$ and $\mathtt{pa}_2(i) = i - 2$, respectively.

> **Mechanism 2 (Copying Head).**
>
> - **Layer 1 Attention: Copying Data to Label Positions.** The QK circuit $W_{\mathrm{QK}}^{(1)}$ *copies $x_i$* at position $(2i - 1)$ *to $y_i$ positions* $2i$ by assigning $W_{\mathrm{QK},\mathrm{pos}}^{(1)}(2i, 2i - 1) \rightarrow +\infty$.
> - **Layer 2 Attention: Data Aggregation.** Layer 2 attention aggregates information from context positions to the query position, preparing for the final prediction.
> - **FFN: Nonlinearity Adaptation.** FFN at layer 1 and layer 2 fit the task-specific nonlinear link function and filter out noises from residual link with their powerful nonlinearity expressiveness.

**Mechanism Verification.** We focus on copying head mechanism at attention layer 1 and consider setup $\mathtt{Lin}$ for verification. To be more specific, we visualize $W_{\mathrm{QK},\mathrm{pos}}^{(1)}$ of $\mathtt{Lin}$ in Figure 3 (i) and observe the attention weight of entry $W_{\mathrm{QK},\mathrm{pos}}^{(1)}(2i, 2i - 1)$ approaches 40. It is also notable that $\mathtt{Quad}$ and $\mathtt{Cube}$ pretraining checkpoints fail to learn the copying head, while for their SFT conterparts $\mathtt{Lin} \rightarrow \mathtt{Quad}$ and $\mathtt{Lin} \rightarrow \mathtt{Cube}$, it is a different story.

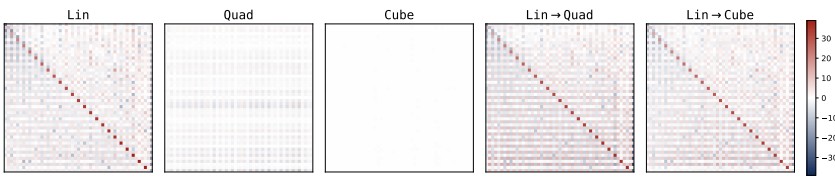

Figure 3: Visualization of $W_{\mathrm{QK},\mathrm{pos}}^{(1)}$ for ICL Regression checkpoints. **(i) Left 3 Heatmaps: Pretraining Checkpoints.** $\mathtt{Lin}$ learns copying head through pretraining with attention weights of entries $W_{\mathrm{QK},\mathrm{pos}}^{(1)}(2i, 2i - 1)$ approach 40. $\mathtt{Quad}$ and $\mathtt{Cube}$ checkpoints fail to learn copying head; **(ii) Right 2 Heatmaps: SFT Checkpoints.** Through SFT from $\mathtt{Lin}$ to high-order polynomials, $\mathtt{Lin} \rightarrow \mathtt{Quad}$ and $\mathtt{Lin} \rightarrow \mathtt{Cube}$ maintain the copying head from $\mathtt{Lin}$ checkpoint, fostering generalization to harder subtasks.

## 4.2 SFT Stage: The Learned Copying Head serves as Deterministic Curriculum for High-Order Polynomial Regression Generalization

We conduct two SFT experiments: Lin → Quad and Lin → Cube, whose results are compared to the controlled group Quad and Cube, respectively. We apply MSE loss as the metric of generalization efficacy, which is visualized in Figure 4. The results demonstrate that all SFT checkpoints generalize to higher polynomial subtasks strongly better than pretraining. In terms of generalization mechanism, we summarize the following Takeaway:

> **Takeaway 3: Data Copying Head serves as Deterministic Curriculum for Harder High-Order Polynomial Regression Generalization.** Data Copying Head circuit remains unchanged throughout SFT, which foster nonlinear adaptation of layer 2 FFN to the assigned high-order polynomial subtask strongly better than pure pretraining.

**Experiment Design.** We make comparisons between 2 SFT checkpoints {Lin → Quad, Lin → Cube} and 2 pretraining checkpoints {Quad, Cube} around 3 claims: (i) **Direct Pretraining High-Order Polynomials Fails to Learn Copying Head.** We visualize $W_{\text{QK,pos}}^{(1)}$ for Quad and Cube to confirm they fails to learn the copying head; (ii) **SFT checkpoints Maintains the Copying Head.** We visualize $W_{\text{QK,pos}}^{(1)}$ for Lin → Quad and Lin → Cube to confirm the copying head is unchanged; (iii) **FFN Adapts to Subtask Nonlinearity via SFT.** Throughout SFT, we plot the Frobenius norm (F-norm) of layer 2 FFN for each setup and conduct 2 comparisons: a) between Lin → Quad and Quad, and b) between Lin → Cube and Cube. We monitor the change of F-norm to evaluate the nonlinearity that FFN adapts to. A far deviation from the random initialized value indicates stronger nonlinearity.

**Experiment Results.** For claim (i), we observe from Figure 3 (i) that the pretraining checkpoints Quad and Cube fail to capture data-label positional dependency to formulate a data copying head. For claim (ii), we observe from Figure 3 (ii) that the SFT checkpoints Lin → Quad and Lin → Cube maintain the copying head from Lin pretraining checkpoint. For claim (iii), we observe in Figure 4 (ii) that the F-norm of Quad and Cube are close to the random initialized value ($< 17\%$), while that of Lin → Quad and Lin → Cube significantly deviate from it ($> 50\%$), demonstrating strong nonlinearity adaptations.

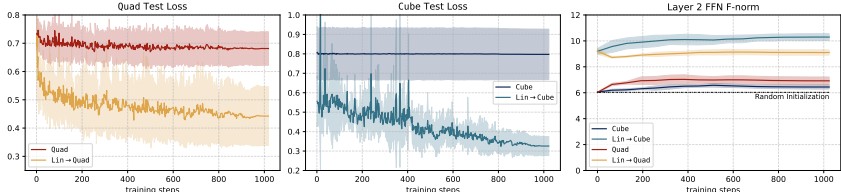

Figure 4: Generalization evaluations of ICL Regression SFT, where formal definitions are deferred to Appendix C. **(i) Left and Middle: Test Loss of Quad and Cube.** Pretraining checkpoints Quad and Cube remain their loss plateau, while SFT checkpoints Lin → Quad and Lin → Cube drop within 50 ($< 5\%$ total) training steps; **(iii) Right: F-norm of Layer 2 FFN.** F-norm of pretraining checkpoints Quad and Cube remain close ($< 17\%$) to random initialization value, while that of SFT checkpoints Lin → Quad and Lin → Cube deviates far ($> 50\%$) from it, revealing successful nonlinearity adaptations.

## 5 Conclusion

We conducted a mechanistic study of how in-context learning (ICL) transfers across tasks. Experiments on Markov chains and regression show that stable structures such as induction circuits and data-copying heads support generalization, while task-specific components adapt to new expertise. These mechanisms explain the efficiency gains from curriculum-style training and point to task sequencing based on mechanistic similarity as a principled strategy for more efficient language model training.

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

## A  HYPERPARAMETER SETTING

Training uses an AdamW optimizer with initial learning rate $\eta = 5 \times 10^{-4}$, cosine annealing, and batch size $n_{\text{batch}} = 64$. A training step contains 128 batches. For ICL Markov Chain task, we train for 1024 steps under 2-gram setting, and train for 2048 steps under 3-gram setting. For ICL Regression task, we train for 1024 steps for all settings.

## B  ATTENTION LAYERS AND FFN LAYERS.

First, we define the causal attention layer, which allows tokens to gather information from previous tokens in the sequence.

**Definition B.1** (Attention layer). *A causal attention layer with $H$ heads is denoted as $\texttt{Attn}_\theta(\cdot)$ parameterized by $\theta$, where $\theta = \{Q_h, K_h, V_h, O_h\}_{h=1}^{H} \subset \mathbb{R}^{D \times D}$. On any input sequence $X \in \mathbb{R}^{L \times D}$,*

$$\texttt{Attn}_\theta(X) := X + \sum_{h=1}^{H} \texttt{smax} \circ \texttt{msk}\big((XQ_h)(XK_h)^\top\big) \cdot XV_h O_h \in \mathbb{R}^{L \times d}, \tag{B.1}$$

*here $\texttt{smax}(\cdot)$ denotes the column-wise softmax operation and $\texttt{msk} : \mathbb{R}^{(d+1) \times (L+1)} \mapsto \mathbb{R}^{(d+1) \times (L+1)}$ denotes the element-wise causal mask and its $(i, j)$-th entry is $\texttt{msk}(\cdot)_{ij} = \text{Id} \cdot \mathbb{1}(i < j) - \infty \cdot \mathbb{1}(i \geq j)$.*

Next, the attention output is processed by a position-wise FFN to add computational depth.

**Definition B.2** (FFN layer). *An FFN layer is denoted as $\texttt{FFN}_\theta(\cdot)$ parameterized by $\theta$, where $\theta = \{W_1, W_2\} \subset \mathbb{R}^{D \times D}$. Let $\sigma$ denote the ReLU function. On any input sequence $X \in \mathbb{R}^{L \times D}$,*

$$\texttt{FFN}_\theta(X) := X + \sigma(XW_1)W_2. \tag{B.2}$$

## C  EVALUATION METRICS

**ICL Markov Chain.**  We use test loss $\mathcal{L}_{\text{CE}}(\theta)$, test accuracy $\texttt{acc}(\theta)$, and test KL-divergence $\texttt{KL}(\theta)$ to evaluate the generalization efficacy of the model, which are defined as follows:

$$\mathcal{L}_{\text{CE}}(\theta) = \mathbb{E}_{\pi \sim \mathsf{P}_\pi, \, s_{1:L+n} \sim \mathsf{P}(\cdot \mid \texttt{pa}, \pi)} \big[ - \log \big( \texttt{smax} \circ \texttt{read} \circ \texttt{TF}_\theta \circ \texttt{emb}(s_{1:L+(n-1)})_{s_{L+n}} \big) \big],$$

$$\texttt{KL}(\theta) = \mathbb{E}_{\pi \sim \mathsf{P}_\pi, \, s_{1:L+n} \sim \mathsf{P}(\cdot \mid \texttt{pa}, \pi)} \big[ \texttt{KL}\big( \texttt{smax} \circ \texttt{read} \circ \texttt{TF}_\theta \circ \texttt{emb}(s_{1:L+(n-1)}) \| \hat{\pi}(s_{1:L+(n-1)}) \big) \big],$$

$$\texttt{acc}(\theta) = \mathbb{E}_{\pi \sim \mathsf{P}_\pi, \, s_{1:L+n} \sim \mathsf{P}(\cdot \mid \texttt{pa}, \pi), \, s \sim \texttt{smax} \circ \texttt{read} \circ \texttt{TF}_\theta \circ \texttt{emb}(s_{1:L+(n-1)})} \big[ \mathbb{1}\big( s = s_{L+n} \big) \big],$$

$$\tag{C.1}$$

where $\mathsf{P}(\cdot \mid \texttt{pa}, \pi)$ denotes the test dataset joint distribution of Markov chain trajectory under parent structure $\texttt{pa}$ and transition rule $\pi$, $\hat{\pi}$ is the empirical estimator defined in Eq. (3.1), $\texttt{KL}(\cdot \| \cdot)$ is the Kullback-Leibler divergence, and $\mathbb{1}(\cdot)$ is the indicator function. Here, $\texttt{emb}$ and $\texttt{read}$ denote the data embedding and readout operators.

**ICL Regression.**  We use test loss to evaluate the generalization efficacy of model, and use F-norm to evaluate nonlinearity adaption of FFN layers. We define test loss $\mathcal{L}_{\text{MSE}}(\theta)$ below:

$$\mathcal{L}_{\text{MSE}}(\theta) = \mathbb{E}_{\beta \sim \mathcal{N}(\mathbf{0}, I), \, (x_i, y_i)_{i=1}^{N} \overset{\text{i.i.d}}{\sim} \mathsf{P}(\cdot \mid \varsigma, \beta)} \Big[ \big( \texttt{read} \circ \texttt{TF}_\theta \circ \texttt{emb}(x_{1:N+1}, y_{1:N}) - y_{N+1} \big)^2 \Big], \tag{C.2}$$

where $\mathsf{P}(\cdot \mid \varsigma, \beta)$ denotes the test dataset joint distribution of any input-output pair $(x_i, y_i)$ under link function $\varsigma$ and parameter vector $\beta$. Here, $\texttt{emb}$ and $\texttt{read}$ denote the data embedding and readout operators.

For FFN parameters at layer $\ell$: $\theta_{\text{FFN}}^{(\ell)} = \{W_1^{(\ell)}, W_2^{(\ell)}\}$, we define its F-norm $\mathcal{F}(\theta^{(\ell)})_{\text{FFN}}$ below:

$$\mathcal{F}(\theta_{\text{FFN}}^{(\ell)}) = \sum_{i,j} \Big( W_1^{(\ell)}(i, j) \Big)^2 + \Big( W_2^{(\ell)}(i, j) \Big)^2 \tag{C.3}$$

# D  SUPPLEMENTARY SETUP

## D.1  DETAILS OF PARENT STRUCTURE `free`

We consider only one subtask setup considering parent structure `free`: $(2, \texttt{free})$, whose parent structure $\texttt{pa}_1$ is randomly sampled, but fixed after sampling and remains unchanged during training. The parent structure $\texttt{pa}_1$ of $(2, \texttt{free})$ for the consecutive $40$ non-root nodes is defined as follows:

$$
\begin{aligned}
\texttt{pa}_1(2:41) = (&1, 2, 3, 1, 4, 4, 7, 8, \\
&2, 4, 10, 3, 4, 6, 14, 9, \\
&4, 1, 19, 14, 21, 4, 17, 15, \\
&25, 10, 8, 18, 5, 1, 21, 21, \\
&9, 17, 11, 34, 30, 20, 18, 9)
\end{aligned}
\tag{D.1}
$$

where token position $1$ is a root node. The corresponding causal matrix $M^1$ is visualized in Figure 1 (i).