# OpenReview forum: "Mechanistic Interpretability of In-Context Learning Generalization through Structured Task Curriculum"
_ICLR.cc/2026/Conference — Submitted to ICLR 2026_

### Official Review · Reviewer_5mQG · 2025-10-27

**Soundness:** 3
**Presentation:** 3
**Contribution:** 2
**Rating:** 4
**Confidence:** 4

**Summary:**

This paper is an empirical study showing cirriculum training works, especially when needed mechanisms are shared between training stages. It provides detailed mechanistic interpretation of small transformers trained on two types of synthetic ICL tasks: ICL markov chain, where the transition matrices are learned from the context (they are different for different sequences), and ICL regression where coefficients are learned from the context (also sequence specific).  It shows detailed evidence of how the shared circuits are maintained between training stages (if they are needed for both stages), and how the already learned circuit in first stage accelerate learning in second stage. Especially in non-linear regression tasks, where a successful circuit requires multiple chained sub-circuits, naive training failed. But cirriculum training help the model learn part of the circuit in first stage, this foundation makes it much easier to learn other parts in the second stage.

**Strengths:**

- The paper shows concrete examples of why and how cirriculum training helps. It provides detailed and understandable analysis on underlying mechanisms. The results highlight the importance of shared or common circuits between training stages.
- The paper designs families of synthetic tasks, enabling easy and systematic control of the amount of shared structure between tasks/training stages.

**Weaknesses:**

- The experiments are not extensive enough. Currently there are only a few tasks, (2, diag), (3, diag), (2, free), and Lin, Quad, Cube. There is also only one model architecture tested in each setting. At least authors can try more configurations like (3, diag) -> (4, diag), and draw conclusion based on more evidence. It will be even better to design new family of tasks. This will make the results more convincing.

- The model architecture used in the paper are transformers with at most two layers (sometimes without MLP layers). While it makes it easy to interpret the model, it also makes the setting looks toy-ish, raising concerns about applicability of the conclusion to more realistic setting.

- It shows that task sequencing based on mechanistic similarity can be a good strategy for efficient training, but it requires anticipating the underlying mechanisms of the sequence of tasks. In other words, we need to successfully guess what the model would learn in different tasks, and what is needed in the final circuit. These should be concrete mechanisms like copying, induction head etc, instead of coarse-grained concepts such as linguistic knowledge. It can be hard to anticipate detailed mechanism for more realistic and complex tasks.

**Questions:**

- Suggestion: The terminology used in the paper, “pretraining” and “fine-tuning” are somewhat misleading. In the paper the two training stages are of similar scale, instead of a large scale pretraining and small scale fine-tuning (i.e., substantially differ in terms of data amount and diversity, and training steps). I would simply call them training stage 1 and 2.

---

### Official Review · Reviewer_5nJy · 2025-10-28

**Soundness:** 1
**Presentation:** 2
**Contribution:** 1
**Rating:** 2
**Confidence:** 4

**Summary:**

This paper aims to aims to mechanistically study generalization mechanisms of in-context learning (ICL). Specifically, it considers two commonly used testbeds for studying ICL in transformers trained from scratch, namely (first-order) Markov chains and linear regression, and aims to extend the mechanistic analyses done in previous works in these settings, to the learning of other tasks that may reuse the learned structures from pretraining, when fine-tuning the transformer on higher-order Markov chains, or nonlinear functions on top of the linear regression, respectively. It shows that this type of curriculum learning leads to more efficient learning of the more complex version of the task, as compared to training the model from scratch. As an explanation, it shows that certain learned structures, such as induction circuit for Markov chains, or copying head for regression remain in tact which helps generalization.

**Strengths:**

The main strength of the paper is that the problem it seeks to address is important and interesting — using a mechanistic lens to understand how curriculum learning can help in ICL settings.

However, there are several weaknesses as discussed in the next section.

**Weaknesses:**

I have several concerns with the paper, spanning the discussion on related work, the experimental setup, and the experimental evidence, as detailed below. Other suggestions are deferred to the next section.

 ### 1. The paper is missing citations to several related works on ICL in transformers.

There is a large body of work studying ICL in transformers trained from scratch on different function families and mechanistic analyses. The paper is missing citation to the seminal work in this direction [1], as well as several other relevant works that study ICL on linear regression, e.g., [2-6] to list a few, and Markov chains [7-8]. Additionally, [9] is very relevant as it also studies both testbeds and considers training on mixture of first- and higher-order Markov chains, and should be discussed for a similar task family (curriculum instead of mixture) considered in the paper. The authors should cite these papers and also conduct a thorough literature search to include all relevant related work.

### 2. The paper misrepresents some of its contributions pertaining to the experimental results for curriculum learning.

In Section 1, the paper states two observations related to curriculum learning as their contributions, which is not accurate. Also, there is no supporting evidence for one of the contributions. Specifically,

- The second contribution states that the paper finds that pretraining reduces data requirements by upto 10 times — this has been seen in other settings in prior work and does not constitute a contribution.
- It also states that the paper shows that “transfer efficiency correlates with formal task similarity metrics”, which is not supported by any evidence in the paper. There is no task similarity metric introduced in the paper.
- The third contribution states that “…training on diverse tasks simultaneously may be suboptimal. Our results advocate for structured curricula…“. Again, this is not a novel finding and authors should appropriately contextualize their contributions with respect to prior work.

### 3. The way the experiments are set up or described for the two main settings (Markov chains and linear regression) is not standard. There is no discussion or justification about this difference with prior works.

For Markov chains, prior works like [6, 8] sample each sequence from a Markov chain (or transition matrix). This paper follows a different process and instead has a context subsequence, followed by a query subsequence to evaluate the model’s predictions. Why does the paper not use the standard way?

Similarly, for linear regression, prior works usually construct sequences with alternating data points and labels, followed by the query, $(x_1, y_1, x_2, y_2, \dots, x_N, y_N, x_{N+1})$. The paper instead first constructs $(x_1, \dots, x_N, x_{N+1}, y_1,\dots,y_N)$ and then does a position rearrangement step to arrive at the same sequence. Why not directly write it the standard way?

### 4. The experimental results for the main claims are not convincing.

My main concern is about the experiment probing a (2, free) model (around line 330). The current experiment uses the causal matrix for (3, diag) to replace $W_{QK}^1$ arguing that given this correct causal structure, generalization (when fine-tuning on (3, diag)) should be fast because the layer 2 induction circuit is already learned. However, a more convincing experiment would be to check whether when all other weights of this (2, free) model are frozen, then fine-tuning only the $W_{QK}^1$ layer is sufficient to generalize.

Also there should be more discussion on why the (2, free) → (3, diag) FT does worse than (2, diag) → (3, diag). Is this an optimization issue? Otherwise, this shows that besides the induction circuit the causal structure from PT to FT also matters.

Next, for the regression setting, around line 460, the paper argues that the Frobenius norm of layer 2 FFN captures the learning of the nonlinear structure. While the norm is larger than random initialization, there is not much change during FT. Can the authors clarify how this supports their claim?

**References:**

[1] Garg et al., “What Can Transformers Learn In-Context? A Case Study of Simple Function Classes”, NeurIPS 2022.

[2] von Oswald et al., “Transformers learn in-context by gradient descent”, ICML 2023.

[3] Ahn et al., “Transformers learn to implement preconditioned gradient descent for in-context learning”, NeurIPS 2023.

[4] Raventos et al., “Pretraining task diversity and the emergence of non-bayesian in-context learning for regression”, NeurIPS 2023.

[5] Lin and Lee, “Dual operating modes of in-context learning”, ICML 2024.

[6] Collins et al, “In-Context Learning with Transformers: Softmax Attention Adapts to Function Lipschitzness”, NeurIPS 2024.

[7] Rajaraman et al., “Transformers on Markov data: Constant depth suffices”, NeurIPS 2024.

[8] Park et al., “Competition Dynamics Shape Algorithmic Phases of In-Context Learning”, ICLR 2025.

[9] Deora et al., “In-Context Occam's Razor: How Transformers Prefer Simpler Hypotheses on the Fly”, COLM 2025.

**Questions:**

Please see the weaknesses section for the main concerns and questions.

Other suggestions about improving the writing are as follows.

1. The abstract can be refined (particularly the first half, where multi-task learning could potentially be misconstrued as training on task mixtures). ‘Curriculum learning’ can be mentioned early on. Quotation marks in line 25 should be fixed.
2. The phrase ‘empirical machine learning community’ in line 51 is not the best choice when referring to works that study transformers in controlled synthetic settings.
3. Missing hyperlinks in the paper organization section; also this section can be removed to add more important things.
4. Edelman et al., 2024 should be discussed in line 127, not 125.
5. The ‘notations’ paragraph should be moved into Section 2.
6. In Section 2.3 the paper uses ‘supervised fine-tuning (SFT)’ to refer to the fine-tuning on the more complex task. This usage is not correct because in practice, SFT is quite different from pre-training. The terms ‘curriculum learning’ or just FT are better suited for the setting of the paper.
7. Line 245 should be supported with a relevant citation.
8. In the last part of Section 3, the phrase ‘deeper generalization’ should be replaced with ‘better generalization’  or ‘improved generalization’.
9. Fig. 2 should be moved to the previous page for better readability.
10. In Mechanism 2 (Copying Head), why is FFN: Nonlinear adaptivity included? It should be mentioned separately.
11. The SFT discussion in Section 4 should be improved, it can be organized like Section 3.

---

### Official Review · Reviewer_GLL5 · 2025-11-01

**Soundness:** 3
**Presentation:** 3
**Contribution:** 3
**Rating:** 4
**Confidence:** 4

**Summary:**

ICL is a powerful emergent property of language models with strong and interesting capacities to generalize. This work investigates the internal dynamics of small transformers as they learn two tasks: Markov chains with varying dependency graphs, and regression tasks with different link functions. For each of these tasks, the different attention circuits are shown, where the MC experiments show that these models are learning the dependency graphs within the matrices of the first layers of the two layer models trained. For the regression experiments, it's seen that when transferring from linear regression to some other link function, a distinct circuit is learned to starting with the different link function/

**Strengths:**

- Transferring between these Markov-style tasks with different dependencies is an interesting and novel idea, especially for the sparser, less structures dependency graphs. It also provides a clear way of controlling the similarity of different tasks by simply embedding the substructure of one within the other (such as with the different `diag` tasks)
- The figures showcase clearly the differences between what is learned by these different models. This is especially true for the regression tasks

**Weaknesses:**

- Some of the writing was unclear. A few things were difficult to decern on an initial skim, such as the meaning of the different task names, which was important for understanding at the beginning (see questions). The abstract also mentions a "formal measure of similarity" between tasks, but this never appears in the body
- The experiments focus on a two-layer, two-head transformer for all experiments. Testing if these same mechanisms are learned when the model is large with significantly greater expressivity (even with a few more layers) remains unexplored
- The experiments also work with one-hot encodings for the positions rather the sinusoidal or some other positional encoding, which decreases the direct applicability to larger models

**Questions:**

- On the first read, the meaning of `diag` and `free` wasn't clear. It seemed like that was going be discussed in the definition of the task in section 2.2.1 rather than the introduction of section 3. If the figure captions could have a brief caption about these meanings, it would help a lot with the first brief skim of this paper
- Can the details of the model architecture also be included in the appendix with the other hyperparameters? Searching for this took some time, especially since the same doesn't seem to be listed for the regression experiments.

- What are these formal measures of similarity alluded to in the abstract? For the MC experiments, are they supposed to be the count of common dependencies between the two tasks?

- Were there any experiments on (2, free) to (3, free), either with the latter containing the former as a substructure or otherwise? This seems to be an interesting experiment that would validate the transferability based on common substructure as seen with (2, diag) to (3, diag)

---

### Official Review · Reviewer_mj1n · 2025-11-01

**Soundness:** 2
**Presentation:** 3
**Contribution:** 2
**Rating:** 2
**Confidence:** 4

**Summary:**

The paper studies how in-context learning (ICL) mechanisms transfer from simpler pretraining tasks to more complex fine-tuning tasks. The authors consider two settings: (i) Markov chains, pretraining on lower-order dependencies and fine-tuning on higher-order ones, and (ii) regression, pretraining on linear functions and fine-tuning on quadratic functions. They analyze which weights and circuit-like structures persist from pretraining and which adapt during fine-tuning.

**Strengths:**

The problem is interesting and timely: understanding mechanistic transfer in ICL across task complexity.

The paper uses a structured synthetic testbed for this study, which is a reasonable and controllable testbed for such an investigation.

The paper also does a decent job of organizing the results coherently.

**Weaknesses:**

The paper contribution feels marginal relative to prior mechanistic ICL work, and several claims are stronger than the provided evidence or lack supporting evidence. Also, some empirical findings appear at odds with prior results, but the paper does not explain these discrepancies. I try to list some of these issues below:

1. Markov chain (n-gram) mechanism (Sec. 3.1): The described mechanism is same as prior results on how transformers learn Markov structure. The only difference in data setup, which is the parent tokens not being the most recent $n-1$ tokens, reads as a small variation rather than a conceptual advance.

2. The SFT mechanism (Sec 3.2): Claim (i) (paragraph starting at line 329), “Layer 2 induction circuit remains unchanged,” is weakly supported. First, there is no fair baseline for comparing convergence speed in the experiments of this section. A proper comparison would be an experiment where the Layer-1 weights are fixed as in the described setup ((2, free) probing case) but Layer-2 weights are initialized randomly. Second, even with such a baseline, faster convergence when fixing Layer-1 attention to the (3, diag) Markov structure does not, on its own, demonstrate that the layer-2 **circuit** remains unchanged throughout training.

3. Sec 4.1 claims (paragraph starting at line 446 and Mechanism 2 on page 8):

    (a) Claim (iii): Tracking FFN weight norms is not sufficient to support “FFNs adapt to the non-linearity of the link function.” Why should distance from random init quantify the degree of non-linearity?

    (b) Claim (i) and thus (ii): The reported failure to learn non-linear functions like Quadratic during direct pretraining conflicts with prior work that has trained models on such ICL tasks, e.g., [1, 2]. This at least needs a careful comparison and discussion.

4. Some claims in the Contributions section and throughout the paper are not well-supported. For instance: The paper mentions “quantitative gains from curriculum learning,” but these results are not presented or discussed clearly. The abstract and conclusion refer to a “formal similarity measure” between tasks that explains how ICL transfer occurs, yet no such formal measure is defined. The contribution also claims that “training on diverse tasks is suboptimal,” but no experiments test pretraining on diverse task sets. All pretraining is performed on a single simple or complex task. Further, the paper states that it provides “concrete recommendations for task ordering based on mechanistic similarity rather than surface-level task categories,” but these *concrete* recommendations are not discussed anywhere.


[1] Kim J., Kwon S., Choi J. Y., Park J., Cho J., Lee J. D., Ryu E. K. Task diversity shortens the ICL plateau.

[2] Garg S., Tsipras D., Liang P. S., Valiant G. What can transformers learn in-context? A case study of simple function classes.

**Questions:**

Other than the points discussed above, I have a few additional minor questions:

1. Notation: Should the link function $\zeta$ be $\mathbb{R}\to\mathbb{R}$?
Also, using $x$ for both the full sequence and the input to the link function $x_{1:N}$ is confusing.

2. Why do you report test accuracy for the Markov task? Since the target is a probability distribution over all tokens, KL and test loss seem more appropriate. What extra insight does accuracy provide?

3. Line 369: What is the position-embedding domain of $W_{KQ}$.

---

### Meta-Review · Area_Chair_9TvT · 2026-01-09

**Summary:**

###### Reviewer mj1n

(1) The mechanism of in-context learning of Markov chains has been discussed in prior work, and only slightly modified here; (2) Several claims made on the basis of empirical results are only weakly supported; (3) Several results, concepts, and implications are referred to, but not clearly defined or explained.

###### Reviewer GLL5

(1) Like Reviewer mj1n (3) above: "Several results, concepts, and implications are referred to, but not clearly defined or explained"; (2) Generalizability of the results to more than 2-layer, 2-head transformers, and to other types of positional encodings, is not established.

###### Reviewer 5nJy

As the reviewer themselves writes: (1) The paper is missing citations to several related works on ICL in transformers; (2) The paper misrepresents some of its contributions pertaining to the experimental results for curriculum learning; (3) The way the experiments are set up or described for the two main settings (Markov chains and linear regression) is not standard. There is no discussion or justification about this difference with prior works; (4) The experimental results for the main claims are not convincing.

###### Reviewer 5mQG

(1) Generalizability of the results to more than the few synthetic Markov chains and the one model architecture here, and to other types of positional encodings, is not established; (2) Like Reviewer GLL5 (2): generalizability of the results to more than 2-layer, 2-head transformers, and to other types of positional encodings, is not established; (3) The mechanistic explanations provided to account for ICL behavior are not at the right level of abstraction: copying and induction heads should be preferred.

**Reviewer Concerns:**

The authors did not post a response, or at least did not post one with the appropriate visibility settings, and so I am unable to assess whether concerns would be addressed by a rebuttal.

I will attempt to make my own assessment. The main set of concerns surround the fact that several claims in the paper are only weakly supported by the evidence provided: Reviewer mj1n (2) and Reviewer 5nJy (4). This is a critical flaw that I assess would need a major revision to address. More minor concerns around the relationship to prior work from Reviewer mj1n (1) and Reviewer 5nJy (2), clarity from Reviewer mj1n (3) and Reviewer GLL5 (1), and around generalizability of the results to other settings from Reviewer GLL5 (2), Reviewer 5nJy (3), and Reviewer 5mQG (1; 2) may have been addressable in a minor revision.

Reviewer 5mQG (3) seems to be a matter of preference and is not clearly articulated so I have disregarded it.

**Reviewer Scores:**

Since half of the reviewers had major concerns, and all reviewers had several minor concerns, I do not anticipate that this paper would receive majority positive scores after discusssion. As such, this paper does not yet seem ready for acceptance at ICLR.

---

### Decision · Program_Chairs · 2026-01-26

Reject